# Sexual Dysfunction in Young Women with Type 1 Diabetes

**DOI:** 10.3390/ijerph17124468

**Published:** 2020-06-22

**Authors:** Edyta Cichocka, Michał Jagusiewicz, Janusz Gumprecht

**Affiliations:** 1Department of Internal Medicine, Diabetology and Nephrology, Faculty of Medical Sciences in Zabrze, Medical University of Silesia, 40-055 Katowice, Poland; jgumprecht@sum.edu.pl; 2Student Scientific Group at the Department of Internal Medicine, Diabetology and Nephrology, Faculty of Medical Sciences in Zabrze, Medical University of Silesia, 40-055 Katowice, Poland; michal.jagu@gmail.com

**Keywords:** sexual dysfunction, type 1 diabetes, young woman, the acceptance of Illness

## Abstract

Introduction: Sexual dysfunctions (SD) are chronic complications that can develop due to vascular complications or autonomic neuropathy. Additionally, such complications can be of hormonal, infectious or psychogenic etiology. Objectives: The aim of study was to assess the sexual function and acceptance of the chronic disease in young sexually active women with type 1 diabetes (T1DM). Materials and methods: A total of 169 female patients with T1DM completed two standardized questionnaires, the Female Sexual Function Index (FSFI) and the Acceptance of Illness Scale (AIS). Other medical data were collected from medical history. Results: The mean FSFI score was 27.96 ± 5.00, and the mean AIS score was 29.67 ± 8.28. The score < 26 points in FSFI was obtained by 28.7% of patients. Analysis of correlation between the FSFI and the AIS showed that the higher the score on the FSFI, the higher the score on the AIS. Patients who underwent regular physical activity (55%) had a significantly higher acceptance of the disease (*p* = 0.0026) and reported painful intercourse significantly less frequently (*p* = 0.01). The value of HbA1c in the study group was 7.31 ± 1.25%. Patients with poorer glycemic control (HbA1c > 8%) obtained significantly lower scores on the FSFI (*p* = 0.03), whereas no differences were found on the AIS. Diabetes-related complications were observed in 25.5% of patients. The presence of chronic complications did not affect the results of the FSFI or the AIS. Patients with diabetes and hypertension had poorer functioning in the sexual sphere and had significantly lower scores on the FSFI. Past or present history of depression was reported by 36% of patients and also negatively affected acceptance of diabetes (*p* = 0.0015). Patients who reported recurrent urinary tract infections (17%) achieved significantly lower scores on the FSFI (*p* = 0.03) and showed that sex-related pain was significantly more prevalent (*p* = 0.02). In the case of the statement related to the embarrassment of people around the patient due to diabetes, patients with lower scores complained of SD significantly more often (*p* = 0.0033). Past deliveries, the type of labor, the use of contraceptives or the number of sexual partners had no influence on the overall assessment in both scales. However, in terms of desire, women who had delivered obtained higher scores (*p* = 0.0021). Conclusion: SD in women with T1DM may result from diabetes-related complications, hormonal disorders or recurrent genital or urinary tract infections. However, they usually have a psychological basis due to the lack of acceptance of the problems related to the treatment of diabetes.

## 1. Introduction 

In recent years, an increase in the incidence of autoimmune diseases has been observed. Type 1 diabetes is one of such diseases, and it is always associated with serious psychological problems for patients and their families due to the introduction of many changes in their daily functioning. Necessary changes in lifestyle and diet as well as self-control or insulin therapy are perceived by patients as a significant decrease in the quality of life. In addition, awareness of potential chronic complications of diabetes also negatively affects all aspects of patients’ lives, including the sexual area [1]. The impact of type 1 diabetes on sexual function has been the subject of many studies. However, it has been assessed more frequently in males [2,3,4].

Sexual dysfunction is one of chronic complications of diabetes that can develop due to vascular complications or autonomic neuropathy. Additionally, such complications can be of hormonal or infectious etiology, which is particularly common among women with diabetes [5,6,7,8]. Psychogenic factors that are secondary to diabetes are also involved. The risk of sexual dysfunction increases particularly in patients characterized by the following factors: lack of acceptance of the disease, negative attitude toward treatment and non-compliance with the recommendations [9].

According to various reports, 37–66% of men with type 1 diabetes are affected by sexual dysfunction, including erectile dysfunction (ED) and premature ejaculation (PE) [10], whereas the percentage among women with type 1 diabetes ranges from 18 to 71% [11]. Sexual dysfunction in women with type 1 diabetes includes insufficient lubrication, lowered libido (decrease in or lack of interest in sex), painful intercourse (dyspareunia, vulvodynia), orgasm disorders and vaginismus [6,7,8,11].

Screening tests, diagnosis, appropriate treatment of sexual disorders and sex education in this group of patients may improve the quality of life and also provide better treatment effect and hence a reduction in diabetic complications.

### Aim of the Study

There have not been many reports that assessed the sexual functioning of women with type 1 diabetes. Therefore, the aim of the study was to assess the sexual function and acceptance of the chronic disease (i.e., diabetes) in young (aged 18–45 years) sexually active women with type 1 diabetes. In our study, we included only patients who had been in a stable relationship for at least six months.

Additionally, an attempt was also made to determine the factors that affect the sexual functioning and acceptance of diabetes. The authors focused on the epidemiological study of the population of Polish young women. Therefore, only a homogeneous group of patients with type 1 diabetes was included in the study.

## 2. Material and Methods

The study was comprised of 169 female patients with type 1 diabetes mellitus (mean age 31 years; q1-26; q3-36) who gave their consent to participate in the study. The participants of the study were recruited from the patients of the Outpatient Diabetes Clinic of the Clinical Hospital No. 1, Zabrze, Poland. The study was a survey, and the approval of the Bioethics Committee was not required. The patients were asked to complete two standardized multiple-choice questionnaires, i.e., the Female Sexual Function Index (FSFI) and the Acceptance of Illness Scale (AIS). Other medical data on diabetes, comorbidities and diabetes-related complications were collected from medical history and medical records during visits to the clinic.

The FSFI is a questionnaire for women that consists of 19 questions whose purpose is a multidimensional assessment of sexual functioning over the 4-week time period. This tool enables differentiation of sexual dysfunctions in six areas (domains): (I) desire, (II) arousal, (III) lubrication, (IV) orgasm, (V) satisfaction and (VI) pain. A total of 6 points can be obtained in each domain, and a total of 36 points in the whole survey.

The Index is a generally available, standardized psychometric tool and is the gold standard in the assessment of sexual dysfunction in women. The higher the score, the better the functioning in a given area. A total equal to or less than 26 points from all domains indicates the presence of clinically significant sexual dysfunction.

The AIS was used after obtaining the consent of the Psychological Test Laboratory of the Polish Psychological Association. It is a scale with eight statements describing the consequences of poor health that are related to illness acceptance, limitation of self-sufficiency, a sense of dependence on others and lowered self-esteem. The greater the acceptance of the disease, the better the adaptation and the lower mental discomfort. Adaptation to the disease is the sum of points ranging from 8 to 40. A low score shows a lack of acceptance and adaptation to the disease and strong mental discomfort. The higher the score, the better the acceptance of one’s medical condition and the fewer negative emotions related to the disease.

The study was conducted between 2018 and 2019. A detailed description of the study population is given in Table 1 and Table 2. Gynecological condition of the studied patients is given in Table 3.

### Statistical Analysis

Statistical analysis was performed using STATISTICA SPSS (IBM, New York, NY, USA) for Windows v. 13 and Microsoft Excel 2015 for MAC (v. 15.14). The data were presented as mean ± SD for variables with normal distribution. However, the data were presented as the median with the lower quartile (q1-25%) and the upper quartile (q3-75%) for the variables with other than normal distribution. The Shapiro–Wilk test was used to assess the normality of the distribution in order to compare the results of the two scales (FSFI and AIS) and their domains. Due to the results presented with the ordinal scale when two independent variables were compared, the Whitney U–Mann test with the correction for tied ranks was used, which was applied due to the multiple occurrence of the same rank values of the variables for each scale. Values of *p* < 0.05 were considered statistically significant.

## 3. Results

Patients obtained the mean FSFI score of 27.96 ± 5.00 (the results of individual domains are given in Table 2) and the mean AIS score of 29.67 ± 8.28. The score <26 points was obtained by 28.7% of patients (*n* = 44). Analysis of the correlation between the FSFI and the AIS showed that the higher the score on the FSFI, the higher the score on the AIS (Figure 1).

When the correlation was analyzed between FSFI and AIS as well as the individual parameters, patients who underwent regular physical activity (55%) had a significantly higher acceptance of the disease (33.5 vs. 29 points, *p* < 0.0001) and reported painful intercourse significantly less frequently (domain VI; FSFI, *p* = 0.01). The degree of metabolic control defined as the value of HbA1c in the study group was 7.31 ± 1.25%. Multiple insulin injections were used in 47.3% of patients (HbA1c = 7.5 ± 1.53%), and the personal insulin pump was used in 52.7% of patients (HbA1c = 7.1 ± 0.83%). No differences were observed between the groups (*p* = 0.166). When the HbA1c cut-off value was set at 7%, no differences were observed in the FSFI between the groups, and patients with better glycemic control (HbA1c < 7%) had significantly higher scores on the AIS (*p* = 0.04). However, when HbA1c cut-off value was set at 8%, patients with poorer glycemic control (HbA1c > 8%) obtained significantly lower scores on the FSFI (*p* = 0.03), whereas no differences were found on the AIS. Diabetes-related complications were observed in 25.5% of patients and mainly included diabetic retinopathy (12.4%), neuropathy (5.3%), nephropathy (1%) and visible hypertrophy of the subcutaneous tissue (lipohypertrophy; 14.2%). The presence of chronic complications of diabetes with particular emphasis on lipohypertrophy did not affect the results of the FSFI or the AIS. Other conditions reported by patients including hypothyroidism, Hashimoto thyroiditis and arterial hypertension were treated in a standard manner. Only patients with diabetes and arterial hypertension had poorer functioning in terms of the sexual sphere and had significantly lower scores on the FSFI. Mood disorders were defined as depressed mood at the time of the study or in the past. From the group of patients with mood disorders, only those who used antidepressants were selected for further analysis. Past or present history of depression was reported by 16% of patients and also negatively affected acceptance of diabetes (*p* = 0.0015). No relationship was found between depression and sexual functioning (*p* = 0.39) in the group.

Patients who reported recurrent urinary tract infections (UTIs) (17%) achieved significantly lower scores on the FSFI (*p* = 0.03), and a trend toward significance was observed (*p* = 0.06) in the group of concomitant fungal infections (23%). The analysis of individual domains of the FSFI in patients with recurrent UTIs and fungal infections showed that sex-related pain was significantly more prevalent (domain VI; FSFI, *p* = 0.02). Moreover, a clear trend toward worse sexual satisfaction was observed in this group of patients (domain V; FSFI; *p* = 0.06).

When individual items from the AIS were assessed, in the case of the statement related to the embarrassment of people around the patient due to diabetes, patients with lower scores complained of sexual dysfunction significantly more often (lower FSFI; *p* = 0.0033).

Past deliveries, the type of labor, the use of contraceptives or the number of sexual partners had no influence on the overall assessment in both scales. However, in terms of desire (domain I; FSFI), women who had delivered obtained higher scores (*p* = 0.0021), and a trend toward better sexual satisfaction was observed (domain IV; FSFI; *p* = 0.06). The total result and all domains on the FSFI were significantly lower in the group of patients who reported a worse sex life due to the negative impact of diabetes than in the group that did not report any influence of diabetes on sexual function (Table 4). Additionally, acceptance of the disease was at a significantly lower level in this group (Table 4). Positive influence of diabetes on sexual function was reported by 9.5% of patients. In the study, the positive impact of diabetes on sex life was defined as follows: better control and control over one’s health, conscious decision about the selection of a diet and physical activity as the elements of diabetes treatment and better "awareness" of one’s own body. However, no relationship was found in this respect. Neither education nor marital status had an influence on the results obtained on the scales. However, patients who worked professionally obtained higher scores compared to non-workers (trend toward significance—Table 4). We did not observe the impact of BMI > 25 kg/m^2^, smoking, alcohol consumption, duration of diabetes, the type of therapy, i.e., multiple dose injection (MDI) or continuous subcutaneous insulin infusion (CCII), or the use of glucose monitoring systems on the results of the scales. 

## 4. Discussion

The aim of the study was to assess the sexual function and acceptance of the disease in young sexually active women with type 1 diabetes using the standardized tools (i.e., FSFI and AIS). In total, women with type 1 diabetes obtained the mean FSFI score of 27.9 points, whereas a total of ≤26 points indicated the presence of clinically significant sexual dysfunction. It was found that 28.7% of patients obtained <26 points on the FSFI. A similar prevalence of sexual dysfunction was observed in other studies. For instance, Enzil et al. confirmed the presence of sexual disorders in 27% of patients with type 1 diabetes [8], and in the METRO Study, the percentage of women with type 1 diabetes with sexual dysfunction was 20% [6]. A meta-analysis of 26 clinical trials showed that sexual dysfunction was 2.27-fold more prevalent in women with type 1 diabetes compared to the control group [12]. An important limitation of the present study is the fact that it was conducted only on women with type 1 diabetes, and the comparison with the control group was not made. However, as it was stressed in the aim of the study, we wanted to examine a homogeneous group of young patients with type 1 diabetes and to determine the scale of the problem. In the course of further research, we plan to compare the obtained results with the control group of young, healthy and sexually active women.

The result of the AIS in the study population was 29.6 out of 40 points and (as mentioned earlier) the higher the score, the better the acceptance of one’s own condition and the fewer negative emotions associated with the disease. Our study also confirmed that the better the acceptance of the disease, the higher the score on the FSFI, i.e., better sexual functioning among the study population.

There have been no similar studies that examined the association between acceptance of the disease and sexual functioning in patients with type 1 diabetes. The only study was conducted in Poland and assessed the impact of the acceptance of type 2 diabetes on sexual function in men and women. It showed that the higher the declared level of the patient’s acceptance of his or her illness, the lower the intensity of the occurrence of sexual functioning disorders [9].

In our study, patients with better metabolic control (HbA1c < 7%) obtained higher scores on the AIS, which means that they managed better with problems related to diabetes, had better compliance and participation in the therapeutic process. When individual items of the AIS were assessed, it was confirmed that patients who obtained lower scores related to the question on embarrassment of people around the patient due to diabetes presented sexual dysfunction significantly more often. This may result from the perception of the problems related to self-control and insulin therapy in the company of others. Patients with type 1 diabetes experience stigmatization in their environment, and some of them avoid public places due to diabetes [13], which also corresponds to other aspects of life, including the sexual sphere.

The analysis of the study results showed a beneficial effect of physical activity on the acceptance of the disease and significantly better sexual functioning, particularly in terms of painful intercourse (domain VI of the FSFI). A similar relationship was demonstrated by Filip et al. who reported that in a group of physically active patients with type 1 diabetes, higher FSFI scores were obtained compared to a group with lower physical activity, especially in the domains related to lubrication and orgasm [14]. Physical activity has a beneficial effect on the circulatory system and also increases the blood flow through the pelvic organs. Better blood circulation increases lubrication, genital sensitivity to stimulation and arousal in women [15]. Regular physical activity also promotes weight loss and shapes the figure, which is particularly important to women in terms of acceptance of their own appearance. Of note, better self-esteem means a better quality of sex life.

The analysis of the present study showed that patients with type 1 diabetes with poorer metabolic control (HbA1c > 8%) obtained significantly lower scores on the FSFI, as did patients with arterial hypertension. According to the literature data, 42.1% of women with hypertension in the general population complain of sexual dysfunction in almost all domains, including desire, arousal, lubrication, orgasm, satisfaction and pain. A lack of lubrication (resulting in dyspareunia) is a common recurrent problem in women with hypertension. As in the case of diabetes or poorly controlled diabetes, hypertension adversely affects the blood flow during sexual arousal, leading to reduced lubrication and, consequently, dyspareunia [16]. Our study also confirmed that patients with recurrent UTIs and fungal infections of the genital tract (often associated with uncontrolled diabetes or poorer glycemic control) presented sex-related pain (domain VI; FSFI) and lower sexual satisfaction (domain V; FSFI) significantly more often.

The occurrence of depressive disorders in the study population also deserves attention. Past or present depressive disorders occurred in 16% of the study population and resulted in significantly lower acceptance of diabetes (AIS). In addition, patients who reported a negative impact of diabetes on their sex life obtained significantly worse results in all FSFI domains and lower scores on the AIS.

There are not many reports that assess the occurrence of depressive disorders in young women with type 1 diabetes. Of note, the prevalence of such disorders ranges from 14% to 30% [6,17,18]. Depressive disorders seem to be one of the most important risk factors for sexual dysfunction in women with diabetes. They adversely affect the quality of life, the relationship and the perception of one’s own body, generating a vicious cycle that results in the occurrence of sexual dysfunction [4,6,8,19,20].

In this study, the following had no influence on the sexual functioning of patients: education, marital status, the number of sexual partners, past deliveries, the duration of diabetes or the type of therapy (MDI/CCII). Only women who had delivered showed higher desire (domain I; FSFI) and better sexual satisfaction (domain IV) compared to women who had not delivered. Of note, sexual dysfunction is significantly more prevalent in women in the early stage after delivery, which is the result of delivery, lactation period, hormonal changes or sleep deprivation [21]. However, in the later period, women who delivered are more satisfied; they are aware of their femininity, are sexually experienced, and hence probably have better satisfaction from sex.

Another limitation of the study is also the fact that we did not include sexual dysfunctions of the partners of our patients, which could affect the sexual functionally of our subjects. However, the study was conducted on the patients from our center, and we did not have access to their partners. In addition, the FSFI scale that was used in the study is focused on the assessment of subjective feelings of women related to all aspects of sexual functioning (ranging from desire, arousal, lubrication, orgasm and satisfaction to pain), which may be beyond the influence of sexual dysfunctions of their partners.

## 5. Conclusions

Sexual dysfunction in women with type 1 diabetes is an important and unfortunately often neglected problem. It may result from diabetes-related complications, hormonal disorders or recurrent genital or urinary tract infections. However, they usually have a psychological basis due to the lack of acceptance of the problems related to the treatment of diabetes and the need for an active participation in the therapeutic process. Of note, mood and depressive disorders can in themselves be the causes of sexual disorders. More attention should be paid to this aspect at every diabetic’s visit as well as to education and the use of appropriate treatment.

## Figures and Tables

**Figure 1 ijerph-17-04468-f001:**
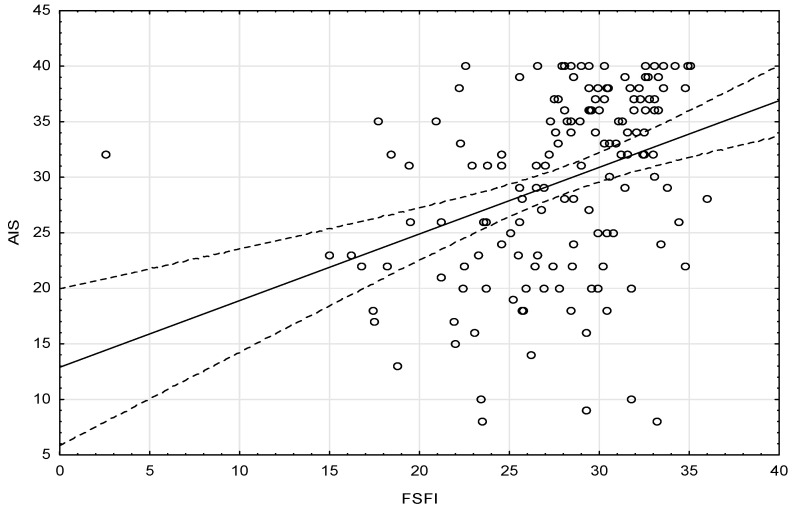
Correlation between Female Sexual Function Index (FSFI) and Acceptance of Illness Scale (AIS).

**Table 1 ijerph-17-04468-t001:** Sociodemographic characteristics of the studied patients.

Variable		*n*	%	Mean	±SD
Education	Primary	4	2.4		
	Secondary	67	39.6		
	Higher	99	58.0		
Professional situation	Worker	124	73.4		
	Non-worker	45	26.6		
Is your financial situation satisfactory?	Yes	119	69.93		
	No	50	29.6		
Marital status	Married	85	50.3		
	Single	74	43.8		
	Divorced	10	5.9		
Stable relationship	Yes	148	87.6		
	No	21	12.4		
Physical activity	Yes	96	55		
	No	76	45		
Frequency of physical activity per week				2.06	0.99
Mean physical activity per week [h]				3.72	1.43
Smoking	Yes	34	20.1		
	No	135	79.5		
Alcohol	None	50	29.6		
	Once per week	27	16.0		
	2–3 times per week	9	5.3		
	Once per month	22	13		
	2–3 times per month	30	17.8		
	Once every 3 months	31	18.3		

**Table 2 ijerph-17-04468-t002:** Clinical and biochemical characteristics of the studied patients.

Variable	*n*	Median	Q1	Q3	Mean	±SD
Age	169	31.00	26.00	36.00		
Weight [kg]	169	65.00	60.00	73.00		
Height [cm]	169	167.00	162.00	170.00		
BMI [kg/m^2^]	169	23.81	21.48	26.45		
Duration of diabetes	169				14.47	8.63
HbA1c [%]	169				7.31	1.25
MDI/CCII [%]	80 (47.3%)/89 (52.7%)					
HbA1c [%]- MDI group	80				7.50	1.53
HbA1c [%]- CCII group	89				7.10	0.83
Continuous glycemic control	35(20.7%)					
Diabetes-related complications:	43(25.5%)					
Diabetic retinopathy Diabetic retinopathy requiring laser therapy	21 (12.4%) 10 (5.9%)					
Nephropathy	2(1.1%)					
Diabetic nephropathy	9(5.3%)					
Visible hypertrophy of the subcutaneous tissue	24(14.2%)					
Other diseases:	86 (50.9%)					
Arterial hypertension	16 (9.4%)					
Hypothyroidism	27(15.9%)					
Hashimoto thyroiditis	29 (17.1%)					
Hyperthyroidism	3 (1.7%)					
Polycystic ovary syndrome (PCOS)	4 (2.3%)					
Mean score of the FSFI	169				27.96	5.00
I—desire					3.94	1.05
II—arousal					4.58	1.06
III—lubrication					4.97	1.05
IV—orgasm					4.62	1.34
V—satisfaction					5.05	0.95
VI—pain					4.81	1.13
Mean score of the AIS	169				29.67	8.29
(Past or present) mood disorders	112 (66.3%)					
(Past or present) depressive disorders	27 (16%)					
Influence of diabetes on sex life: Positive	16 (9.5%)					
Negative	45 (26.6%)					
no influence	108 (63.9%)					

**Table 3 ijerph-17-04468-t003:** Gynecological condition of the studied patients.

Variable		*n*	%	Mean	
Contraception:					
None		54	31.9		
natural methods of contraception (calendar method, interrupted intercourse)		34	20.1		
mechanical contraceptives (condoms, vaginal rings)		70	41.2		
intrauterine contraceptive device		2	1.1		
hormonal contraceptives (pills, patches)		34	20.1		
chemical contraceptives (gels, creams)		5	2.9		
**Deliveries:**	YES	92	54.4		
	NO	77	45.6		
1 delivery		47	51		
2 deliveries		33	35.8		
Natural delivery		38	41.3		
C-section		54	58.7		
mean age of sexual initiation				19.16	3.07
mean number of sexual partners		169		3.69	2.03
mean duration of a stable relationship [years]		148		8.63	6.48
**Gynecological complaints:**		108	63.9		
irregular menstruation (>±7 days)		51	30.1		
recurrent urinary tract infections		29	17.1		
recurrent fungal infections of the genital tract		39	23.0		
vaginal dryness/decreased lubrication		44	26.0		
vaginal pain		15	8.8		
itching in the genital area		36	21.3		

**Table 4 ijerph-17-04468-t004:** Correlation between FSFI, AIS and the studied parameters in type 1 diabetes.

Parameter	Total Score of the FSFI	Total Score of the AIS
education	*p* = 0.16	*p* = 0.36
marital status	*p* = 0.65	*p* = 0.11
work	*p* = 0.06	*p* = 0.05
BMI > 25 kg/m^2^	*p* = 0.54	*p* = 0.74
smoking	*p* = 0.87	*p* = 0.05
alcohol	*p* = 0.69	*p* = 0.33
physical activity	*p* = 0.23	*p* = 0.002
physical activity—VI-FSFI-pain complaints	*p* = 0.01	
disease duration >10 years	*p* = 0.17	*p* = 0.72
type of therapy	*p* = 0.49	*p* = 0.87
use of continuous glucose monitoring system	*p* = 0.47	*p* = 0.38
glycemic control HbA1c > 7%	*p* = 0.15	*p* = 0.04
glycemic control HbA1c > 8%	*p* = 0.03	*p* = 0.88
chronic complications of diabetes	*p* = 0.79	*p* = 0.26
visible hypertrophy of the subcutaneous tissue	*p* = 0.87	*p* = 0.14
Hashimoto thyroiditis	*p* = 0.45	*p* = 0.30
arterial hypertension	*p* = 0.08	*p* = 0.81
past or present history of depression	*p* = 0.39	*p* = 0.0015
gynecological complaints	*p* = 0.03	*p* = 0.87
irregular menstruation	*p* = 0.36	*p* = 0.97
recurrent urinary tract infections	*p* = 0.03	*p* = 0.65
recurrent fungal infections	*p* = 0.06	*p* = 0.76
fungal infections/domain V-FSFI— satisfaction	*p* = 0.06	
fungal infections/domain VI-FSFI-pain	*p* = 0.02	
Item 8 (AIS)—embarrassment of people around the patient due to diabetes	*p* = 0.0033	
mechanical contraception	*p* = 0.72	*p* = 0.40
delivery	*p* = 0.89	*p* = 0.88
delivery/domain I-FSFI—desire	*p* = 0.02	
delivery/domain IV-FSFI—satisfaction	*p* = 0.06	
C-section delivery	*p* = 0.89	*p* = 0.76
number of partners	*p* = 0.62	*p* = 0.55
negative impact of diabetes on sex life	*p* = 0.0001	*p* = 0.00001
negative impact of diabetes on sex life/FSFI domains:		
I-desire	0.0046	
II-arousal	0.0029	
III-lubrication	0.0018	
IV-orgasm	0.0014	
V-satisfaction	0.0005	
VI-pain	0.16

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
