# Peer review of "Sexual Dysfunction in Young Women with Type 1 Diabetes"

_ijerph, 2020, doi:10.3390/ijerph17124468_

Round 1

Reviewer 1 Report

The Authors studied the sexual function and acceptance of this chronic disease in young sexually active women with type 1 diabetes. The female patients completed two standardized questionnaires:  the Female Sexual Function Index (FSFI) and the Acceptance of Illness Scale (AIS).  They concluded that the Sexual Dysfunction in these women may depend on diabetes-related complications, hormonal disorders and or recurrent genital or urinary tract infections. However, there is also a psychological factor due to the lack of acceptance of the diabetic condition.

A major limitation of this study is that the control group is missing.  Therefore, this must be underlined and explained in the discussion.

There are also various potentially confounding aspects in this study:

  1. a) Only diabetic women with a stable relationship of at least 6-12 months should be considered for the study;
  2. b) In addition, no mention is made of any sexual disorders affecting the partner;
  3. c) Table 3 shows:

-"Other diseases": it is necessary to specify if these women are under pharmacological treatment for these pathologies;

-Past or present mood disorders are reported in 66.3% of subjects; there is not the correlation study in  Table 4; moreover,  it is not specified if psycotropic drugs are used.

-Past or present depressive disorders are reported in 16% of women, but it is not specified if psycotropic drugs are used.

It should also be stressed in the discussion that mood and depressive  disorders can in themselves be the cause of sexual dysfunction.

Author Response

Dear Reviewer

Thank you very much for your revision. We truly appreciate your work to improve the manuscript. Please find below the responses to your questions.

  1. A major limitation of this study is that the control group is missing.  Therefore, this must be underlined and explained in the discussion.

An important limitation of the study is the fact that it was conducted only on women with type 1 diabetes and the comparison with the control group was not made. However, as it was stressed in the aim of the study, we wanted to examine a homogeneous group of young patients with type 1 diabetes and to determine the scale of the problem. In the course of further research, we plan to compare the obtained results with the control group of young, healthy and sexually active women.

  1. Only diabetic women with a stable relationship of at least 6-12 months should be considered for the study;

In our study, we included only patients who had been in a stable relationship for at least six months.

  1. In addition, no mention is made of any sexual disorders affecting the partner;

Another limitation of the study is also the fact that we did not include sexual dysfunctions of the partners of our patients, which could affect the sexual functionally of our subjects However, the study was conducted on the patients from our center and we did not have access to their partners. In addition, the FSFI scale that was used in the study is focused on the assessment of subjective feelings of women related to all aspects of sexual functioning (ranging from desire, excitement, lubrication, orgasm and satisfaction to pain), which may be beyond the influence of sexual dysfunctions of their partners.

  1. Table 3 shows:

"Other diseases": it is necessary to specify if these women are under pharmacological treatment for these pathologies;

Other conditions reported by patients included hypothyroidism, Hashimoto thyroiditis and arterial hypertension treated in a standard manner. As included in the results, patients with arterial hypertension obtained significantly lower levels on the FSFI.

  1. Past or present mood disorders are reported in 66.3% of subjects; there is not the correlation study in  Table 4; moreover,  it is not specified if psycotropic drugs are used.

Mood disorders were defined as depressed mood at the time of the study or in the past. From the group of patients with mood disorders, only those who used antidepressants were selected for further analysis.

  1. Past or present depressive disorders are reported in 16% of women, but it is not specified if psycotropic drugs are used.

The study included only depressive disorders that required the use of antidepressants. However, no relationship was found between depression and sexual functioning (p = 0.39). Lower acceptance of diabetes was found only in patients with depressive disorders (p = 0.0015).

  1. It should also be stressed in the discussion that mood and depressive disorders can in themselves be the cause of sexual dysfunction.

We stressed in the discussion that mood and depressive disorders can in themselves be the causes of sexual disorders.

We truly hope the changes in the manuscript were made according to your suggestions. Thank you for your time and valuable remarks. We highly appreciate your involvement in the review.

On behalf of the authors, I wish to thank you very much.

Dr Edyta Cichocka

Reviewer 2 Report

Dear Authors

The study “Sexual dysfunction in young women with type 1 diabetes” is interesting and very well described and written. However, I have major and minor comments, listed below:

  1. Why author have not included healthy controls in this study, which is a major limitation, and even has not been discussed anywhere in the manuscript.
  2. Throughout the article at many places p-value is mentioned as 0.00, which has to be corrected.
  3. Table 2- Authors reported influence of diabetes on sex life: positive (9.5%), negative (26.65), and no influence (63.95). What is positive data mean? Do authors desires to say that 9.5% type 1 diabetes women have better sex life? Whereas, a majority type 1 women have not influence on sex life even having type 1 diabetes. Please explain this part in text clearly with reasoning.
  4. Table 4- p value is missing for physical activity.

Thanks

Author Response

Dear Reviewer

Thank you very much for your revision. We truly appreciate your work to improve the manuscript. Please find below the responses to your questions.

  1. Why author have not included healthy controls in this study, which is a major limitation, and even has not been discussed anywhere in the manuscript.

The authors focused on the epidemiological study of the population of Polish young women. Therefore a homogeneous group of patients with type 1 diabetes was included in the study.

An important limitation of the study is the fact that it was conducted only on women with type 1 diabetes and the comparison with the control group was not made. However, as it was stressed in the aim of the study, we wanted to examine a homogeneous group of young patients with type 1 diabetes and to determine the scale of the problem. In the course of further research, we plan to compare the obtained results with the control group of young, healthy and sexually active women.

  1. Throughout the article at many places p-value is mentioned as 0.00, which has to be corrected.

p-values was corrected

  1. Table 2- Authors reported influence of diabetes on sex life: positive (9.5%), negative (26.65), and no influence (63.95). What is positive data mean? Do authors desires to say that 9.5% type 1 diabetes women have better sex life? Whereas, a majority type 1 women have not influence on sex life even having type 1 diabetes. Please explain this part in text clearly with reasoning.

In the study, the positive impact of diabetes on sex life was defined in the study as follows:

better control and control over one’s health, conscious decision about the selection of a diet and physical activity as the elements of diabetes treatment, better "awareness" of one’s own body.

  1. Table 4- p value is missing for physical activity.

The missing p-value was added.

I truly hope the changes in the manuscript were made according to your suggestions. Thank you for your time and valuable remarks. We highly appreciate your involvement in the review.

On behalf of the authors, I wish to thank you very much.

Dr Edyta Cichocka

Round 2

Reviewer 1 Report

The authors responded adequately to the questions and modified the text as suggested.

Reviewer 2 Report

Dear Authors,

Thank you for the clarification, and making appropriate correction in the manuscript. 

Thanks